# Evaluating the Reaction to a Complex Rotated Object in the American Quarter Horse (*Equus caballus*)

**DOI:** 10.3390/ani11051383

**Published:** 2021-05-13

**Authors:** Megan Elizabeth Corgan, Temple Grandin, Sarah Matlock

**Affiliations:** Department of Animal Science, Colorado State University, Fort Collins, CO 80523, USA; mecorgan@colostate.edu (M.E.C.); sarah.matlock@colostate.edu (S.M.)

**Keywords:** horse, safety, training, behavior, novel object, habituation

## Abstract

**Simple Summary:**

Horses are prey animals and exhibit behaviors that help them adapt and survive in their environment. These reactions are often referred to as spooking and they have the potential to be dangerous to the horse, handler and rider. Spooking consists of avoidance reactions that include suddenly moving away or running away from the perceived danger. It is dangerous for both riders and horses when a horse suddenly startles. Sometimes horses do this in familiar environments because familiar objects may look different when rotated. The purpose of this study was to determine whether horses that had been habituated to a complex object (children’s playset) would react to the object as novel when rotated 90 degrees. Twenty young horses were led past the playset 15 times by a handler. Next, the rotated group was led past the rotated playset 15 times. Each time the horse was led by the object was counted as a pass. An increasing reactivity scale was used to quantify behavioral responses. Being aware of potential reactions to changes in the orientation of previously familiar objects can help keep the handler safer.

**Abstract:**

It is dangerous for both riders and horses when a horse suddenly startles. Sometimes horses do this in familiar environments because familiar objects may look different when rotated. The purpose of this study was to determine whether horses that had been habituated to a complex object (children’s playset) would react to the object as novel when rotated 90 degrees. Twenty young horses were led past the playset 15 times by a handler. Next, the rotated group was led past the rotated playset 15 times. Each time the horse was led by the object was a pass. The behavioral responses observed and analyzed were ears focused on the object, nostril flares, neck raising, snort, avoid by stopping, avoid by moving feet sideways, and avoid by flight. An increasing reactivity scale was used to quantify behavioral responses. A two-sample *t*-test was performed on the reactivity scores comparing the first pass by the novel object to the first pass by the rotated object. The horses in the rotated group reacted to the rotated orientation similarly to the first exposure (*p* = 0.001, α < 0.05). Being aware of potential reactions to changes in previously familiar environments can help keep the handler safer.

## 1. Introduction

Spooking was associated with 27% of horse accidents [1]. Researchers have described “spooking” as “horse reacted in fear of something, unseating the rider” [1]. Another study showed that injuries to riders were more likely to occur “when the horse behaved in an unexpected manner” [2]. Some of the more severe injuries occur when a rider falls of a horse. Data collected from hospital emergency rooms indicated that falling off a horse resulted in more injuries than injuries that occurred when the rider was not mounted [3]. Another study showed that “danger to the rider increases as the horse’s speed increases” [4]. When riders fall of a horse, their head is more likely to be injured compared to other parts of the body [5]. Riders have reported that sometimes they do not know what caused their horse to spook. Experienced horse trainers know that it is important to habituate young horses to new things in their environment. One study showed that gradually desensitizing horses to stimuli such as a moving white nylon bag had fewer flight responses than horses where it was suddenly introduced [6]. Several studies have also indicated that a sudden stimuli such as an umbrella that opens suddenly can startle a horse [7,8,9,10,11]. A sudden novel stimulus such as a waving flag or a bell will raise a horse’s heart rate [12]. Researchers have also reported the intensity of a horse’s reaction was influenced by horse genetics [13]. Sudden novel stimuli will startle many different species of animals [14]. The dangerous and usually unexpected “spook” reaction is likely to be a startle response. 

A review of the horse behavior literature showed that there are different ways that horses are introduced to new objects. They consist of voluntary approach in an open field, led towards the object, or ridden toward the object [15,16,17]. Open field tests are usually conducted in an arena. In these experiments a free moving horse is allowed to voluntarily approach novel objects [15]. In many of these studies the horse learns to associate a particular novel object with a food reward [18,19]. These studies are extremely useful for the study of horse cognition and learning. However they do not provide information on the conditions that may cause a dangerous “spook” reaction that could seriously injure a rider. One study showed that horses that were more willing to investigate a novel object were able to learn more quickly [20]. There may be differences in a horses reaction to a novel object when the way it is presented is changed. Research has shown that a horses reaction to a novel object is different during a voluntary approach compared to being led or ridden [17]. The authors found that when including a rider in temperament tests and performance tests, results were more reliable and repeatable. When positive reinforcement such as food is used, horses are able to choose the correct object and be presented with a reward. This type of learning and test does not often include a handler and is referred to as discrimination [18].

The purpose of this research was to conduct a study to identify a possible situation where a large complex novel object may have the potential to cause a serious horse accident. The object we chose was a child’s colored plastic playset with a small slide and swing. It is a large object and large objects may be more frightening stimulus compared to smaller objects [21]. The playset is an object that a horse could possibly see during riding near residences. It is known that horses can learn to discriminate between different colors and shapes [18,22]. In our experiment, no attempt was made to control for color or shape, but a playset was chosen that had approximately the same dimensions when it was rotated.

The purpose of this preliminary study was to determine if a habituated horse may react differently when the playset was rotated. Both its shape and color would change. A non-voluntary approach was used. All horses were led at a walk for both habituation and the test after the playset was rotated. For safety reasons, we conducted all experiments at a walk and measured behavioral reactions used by Leiner and Fendt [7]. Their study may provide insights into a situation that might trigger dangerous “spooking”. If the rotated playset is associated with increased mild behavioral reactions when a horse is being led towards it, there is the possibility that it might trigger a more dangerous sudden reaction if a horse and rider was trotting towards it. A high speed study would be too dangerous and would be unethical. This study may help identify one cause of spooking when a rider is not able to identify an obvious possible trigger. It will also provide a starting point for further studies into horse perception.

## 2. Materials and Methods

The sample population consisted of twenty-two 2 and 3-year-old American Quarter Horse horses (15 fillies and seven gelded colts) in a university horse training program. The horses had 4 months of handling training at the time of this study, all trained at the same place. The horses were taught using methods of pressure and release training to halter, lead, lunge, and acclimate to being groomed and handled. Of the twenty-two horses, one posed a safety concern for the research handlers by its continued attempt to pull away and was excluded from the study. Another horse was removed from the study on day 4 for soundness issues. Twenty horses continued through the entire study (*n* = 20). All horses were housed at the Colorado State University Equine Teaching and Research Center (CSU ETRC) in outdoor pens with ad libitum water and access to shelter. Horses were fed a mix of grass and alfalfa hay once per day on a feed bunk.

The test environment was an alley (4.57 meters wide) in an indoor horse barn in front of empty stalls with the doors closed. The barn had concrete flooring and electric lights above the alleyway. The horses were led in through the entrance, walked down the alleyway, past the novel object and led out of the test area through the exit (Figure 1). Two Hero 5 video cameras (GoPro, San Mateo, CA, USA) were placed in the test environment for later observation of behavioral responses.

The novel object was a children’s plastic playset (Little Tikes Hide and Seek Climber and Swing-Brown and Tan) (Table 1). The object was 134.62 l × 132.08 w × 104.14 h cm. This object was used because, in both orientations, its outer dimensions are similar. The playset had a different shape when rotated ninety degrees. Rope halters with 2 m lead ropes were used to lead the young horses past the playset.

On day 1–3 of the study (habituation to the test area) (Table 1), the horses were led through the test area five passes each day without the novel object to habituate the horses to the test area. Each time a horse was led past the object was counted as a pass. The horses were given 15 total passes through the test area based on Christensen et al. [15]. The authors found that horses needed 4–13 exposures before meeting habituation criterion. 

On day 4, the novel object was placed in the test area in the original position. Days 4–6 of the study (habituation to the novel object) consisted of the same procedure for the first three days with the novel object in its original position (Table 1). Each horse passed the original position of the object fifteen times over three days. 

To assess the effect of object rotation on the horses’ behavior, the horses were randomized into a control group and a rotated group. On days 7–9 (test days), the control group had five passes each day through the test area, with the novel object in the original position (Table 1). The control group passed the original position of the object fifteen times over the three days. The rotated group was led through the test environment for five passes each day with the novel object rotated 90 degrees clockwise (Table 1). The rotated group passed the rotated position of the object fifteen times over the three days.

Two handlers were used and both led the horses at a slow walk. Each handler had an equal number of horses randomly assigned from both the control and rotated group. Each horse was led at a walk through the test area by the same handler for the entire study. The handlers wore the same clothes every day (black overalls, jacket, hat and black boots). The handlers were instructed to walk the horse with a lead rope through the center of the alley (1 m away from the object), and move with the horse, only stopping or turning when the horse stopped or turned towards the object. If the horse stopped, the handler waited 3 seconds before gently encouraging the horse forward by walking forward and slightly pulling on the lead rope. To facilitate habituation, if a horse stopped when it was either approaching or passing the novel object, it was allowed to stop for a period of 3 seconds. If the horse did not stop, the handler continued to lead it past the novel object.

The horses’ behavioral responses to the object exposure was analyzed based on the video recordings. One observer recorded eight different behavioral signs during each pass on each day. The behavioral responses recorded were ears focused on the object, nostril flares, neck raising, snorting, avoid stop, avoid side, avoid back, and avoid flight (Table 2). Behavioral responses were adapted from [7].

A reaction scale was created from the behaviors observed on a scale from 0–3 (Table 3). This reaction scale was adapted from Christensen et al. [15] to evaluate reactivity based on behaviors observed in this study. The reactivity scores increase with a bigger response to the object, score 3 showing the biggest response.

To assess whether the horses’ behavior was affected by the object rotation we compared difference in the reaction score per individual horse using a two-sample *t*-test (α < 0.05) (R with R Studio, PBC, Boston, MA, USA). Christensen et al. and Tidyverse (www.tidyverse.org, accessed on 28 March 2021) was used for the figures [15,24,25]. This test was done for each pass 1–15 comparing the corresponding passes from the original position to the rotated position. 

## 3. Results

The control and rotated group showed significant differences between the change in reaction score from the first pass by the novel object to the first pass on the Test days (*t*-test *p*-value = 0.001) (Figure 2). Horses that reacted to the novel object in the Rotated group, reacted similarly on the first pass by the rotated position of the object as they did on the initial pass by the novel object.

Passes 1–4 after rotation in the rotated group showed a significant difference between the two groups change in reaction (*p*-values = 0.001, 0.010, 0.004, 0.001). After pass 4 by the rotated object, there was little significant difference between the rotated and control groups (*p*-values > 0.05). As noted in Table 4, some later passes also showed significant differences in the change in reaction between the two groups (Passes 1–4, 8, 9, 12: *p*-value > 0.05). Figure 2 and Figure 3 show the significance in the change in reactions for the rotated group when the horses were exposed to the rotated object.

## 4. Discussion

When a previously familiar complex novel object is rotated, the rotated object may cause reactions similar to the initial exposure to the novel object. This confirms what handlers and riders have described anecdotally. Understanding horses’ reaction to a rotated object is important for the safety of riders and handlers. If handlers expect horses not to react to subtle changes in a familiar environment, they are less prepared for a horse spooking which could lead to an accident. Additionally, studies have shown that investigative behavior is correlated with learning [20]. Allowing a horse to investigate and become familiar with all orientations of an object can help to avoid spooking. Future studies are needed to evaluate if allowing a horse to fully investigate a novel object will help with habituation and decrease spooking.

As shown in Figure 3, there was a decline in the horses’ reactions with each successive pass by the rotated object. Table 4 shows the significant decrease in the horses’ reaction to the object with each successive pass, for the first four passes. After pass 4, the changes in reactions between the rotated and non-rotated groups seem to be less consistent. This inconsistency in changes in reactions between the two groups shows the unpredictable nature of the horse [26]. Even subtle changes to a familiar object can cause horses to react again. These subtle changes can cause the horse to need more exposure until they are habituated or until no reactions are shown again. Handlers need to be aware of this for safety of themselves and the horses.

This study shows that despite findings from previous research [19], horses may not recognize different orientations of previously familiar objects, when being led by a handler. While assumptions cannot be made about the horse’s recognition of the rotated object from the present study, there is an obvious reaction to the rotated object. This reaction is important to note and important for anyone handling horses to be aware of. There are possible differences in personality and reactivity by breed [27]. The traits that showed the most variability between breeds were excitability and anxiousness [27]. This is thought to be because of the purpose each breed has been bred for over time. For example, Thoroughbreds tend to show a bigger flight response because they are bred to race and are expected to leave the gate quickly [28]. Age has also been shown to have an effect on reactivity. In a study comparing horses ranging in age from two months to two years, younger horses were more reactive [29].

Training methodologies are worth further exploration when researching equine perception of novel objects. Humans can have an impact on how the horse reacts and behaves [16]. There may be a difference between a voluntary approach, as compared to being led by a handler. Marsboll and Christensen [30] and Hartmann et al. [31] found that a familiar handler can have a calming effect on the horses response to a novel object as well as a change in fear response. The present study used unknown handlers to evaluate how horses will react to a rotated object with a handler. Christensen et al. [20] showed that fearfulness was not correlated with learning, whereas investigative behavior was correlated with learning. While their study did not use handlers, the present study allowed some investigative behavior and kept the handler involved, creating a more practical scenario. The present study did not use food or positive reinforcement when evaluating recognition or reactivity, as compared to [19]. Using food as a reinforcement in training is similar to using latency to eat in research. It is important to note, that most trainers do not use food as reinforcement in their training. 

The purpose of this study was to evaluate how horses being led and habituated to a previously familiar complex object would react after it was rotated ninety degrees. This study showed that horses’ reaction to a rotated orientation of a familiar complex object was similar to its reaction when it first saw it.

## 5. Conclusions

Horses may have a greater reaction to new orientations of previously familiar objects. This may cause accidents that lead to injury of the horse or handler. If handlers and riders can be prepared for how a horse may react, they may be able to help reduce risk by adjusting training methods to allow for investigation of all sides of an object. Additionally, while horses show a decrease in reactions to novel objects and novel orientations of familiar objects, there is the possibility for reactions during habituation. Further research needs to be conducted to evaluate how different methods of handling and training affect the horses’ reaction to changes in their environment.

## Figures and Tables

**Figure 1 animals-11-01383-f001:**
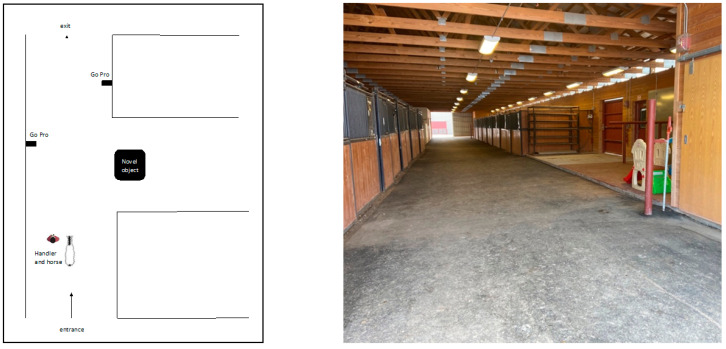
The test area consisted of GoPros (
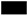
) and the novel object placed during habituation to the novel object and rotation days.

**Figure 2 animals-11-01383-f002:**
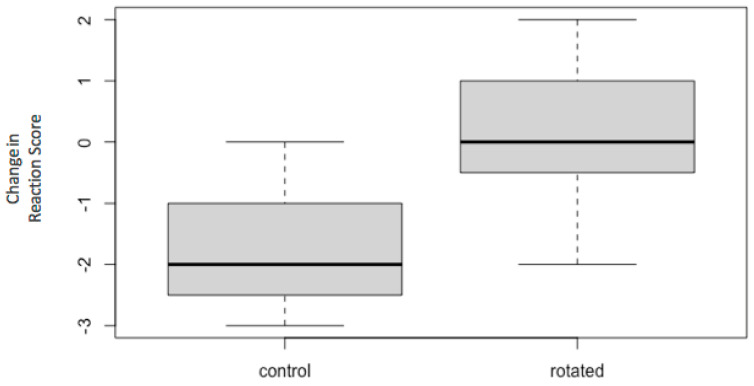
Boxplot of differences in Reaction Score for pass 1 by the novel object to pass 1 by the rotated object. The Control group mean (bold line) showed a decrease in reaction. The Rotated group mean showed no change in reaction. There was a significant difference between the means of the two groups.

**Figure 3 animals-11-01383-f003:**
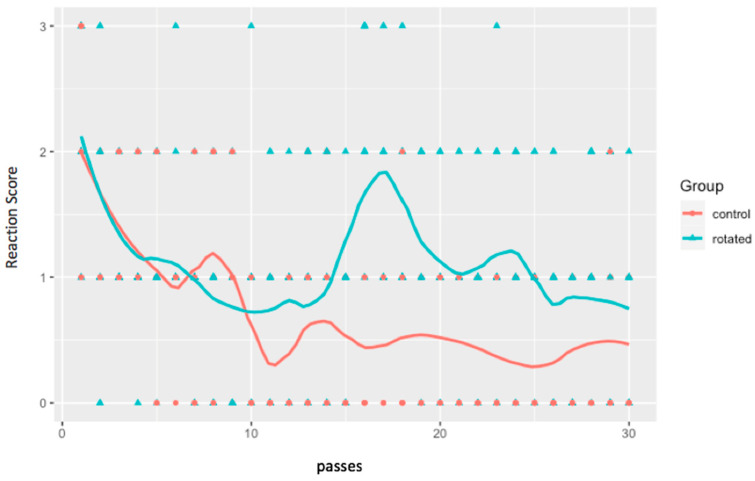
Graph of reaction scores from pass 1–30 for the control and rotated groups. The colored triangles indicate individual horses’ reactivity scores. The colored lines depict mean reaction score by group over passes (control = red, rotated = blue).

**Table 1 animals-11-01383-t001:** Testing Procedure: Outline of testing procedure to provide details of the Control and Rotated group procedures.

Days 1–3	Habituation to test area (novel object absent)	Control and Rotated groups	
		5 passes each day 15 total passes test area (Figure 1) without novel object	
Days 4–6	Habituation to the novel object	Control and Rotated groups	
		5 passes each day 15 total passes test area (Figure 1) with novel object in original position 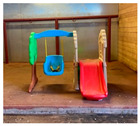	
Days 7–9	Test days	Control group	Rotated group
		5 passes each day test area (Figure 1) with object in original position 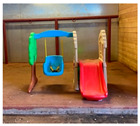	5 passes each day test area (Figure 1) with object in rotated position 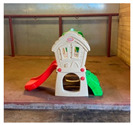

**Table 2 animals-11-01383-t002:** Behavioral responses and definitions used for behavioral analysis.

Definitions of Behavioral Responses	
Behavioral Responses	Definition
ears focused on the object	ears are pointed toward the novel object
nostril flares	nostrils overly expanded (nose elongation)
neck raising	neck raised above normal headset and/or neck muscles tense
snorting	“short powerful exhale” [23]
avoid stop	avoiding the object by stopping, feet stop moving
avoid side	avoiding the object by evasive steps to the side, away from the object
avoid back	avoiding the object by evasive steps backwards, backing up
avoid flight	avoiding the object by jumping away in a sudden movement, feet moving faster a walk

**Table 3 animals-11-01383-t003:** Reaction Scale used to quantify behavioral responses (adapted from Christensen et al. [15]).

Score 0–3	Behavioral Responses Observed
0	No behavioral signs observed
1	Ears focused, nostril flares, and/or neck raising
2	Snorting and/or avoid stop
3	Avoid side, avoid back, avoid flight

**Table 4 animals-11-01383-t004:** Values for differences in reaction score for corresponding passes 1–15 by the novel object to rotated object.

		Control			Rotated		
Pass #	Mean	Min.	Max.	Mean	Min.	Max.	*p*-Value
1	−1.75	−3	0	0.083	−2	2	0.001
2	−0.875	−2	0	0.25	−1	2	0.010
3	−0.875	−2	0	0.167	−1	1	0.004
4	−1	−2	0	0.333	−1	1	0.001
5	−0.375	−2	1	-0.083	−2	0	0.312
6	−0.375	−1	0	-0.25	−1	1	0.719
7	−0.5	−2	0	0.167	−1	1	0.062
8	−1	−2	0	0.333	−1	2	0.005
9	−0.875	−2	0	0.583	−2	2	0.002
10	−0.125	−1	1	0.167	−1	2	0.537
11	0	0	0	0.167	−2	2	0.656
12	0.125	−1	1	-0.333	−1	0	0.010
13	−0.125	−1	1	0.333	−2	1	0.226
14	−0.25	−1	2	0.083	−2	2	0.554
15	0	−1	1	-0.167	−2	2	0.700

## Data Availability

The data presented in this study are openly available in FigShare at [10.6084/m9.figshare.14256512.v1].

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
