# Peer review of "Evaluating the Reaction to a Complex Rotated Object in the American Quarter Horse (Equus caballus)"

_animals, 2021, doi:10.3390/ani11051383_

Round 1
Reviewer 1 Report
This is an interesting work that adds new information regarding reaction of horses, that will help a better and safe management.
Title: Equus should be in uppercase.
Introduction: explains well the topic.
P2L55: in reference 4, the author´s last name should appear more than the reference number.
P2L69: Additionally, A horse´s sudden flight reaction. A should be lowercase.
P2L66-79: some aspects in this paragraph are mentioned several times. For example, the risk of accident with a spooky or scared horse is mentioned five times. In my opinion, that paragraph can be summarized in a central idea that is that reaction in horses or spooky animals can cause accidents and injuries for both riders/handlers and horses, and that those reactions are common in the equine industry.
In addition, in the previous paragraph (L61-63) it was already mentioned that horses led by a handler can change their behavior, similar to L76-77. That idea and study can be mentioned in the second paragraph.
Material and Methods:
Table 2: the title repeats Behavioral signs (P4L147)
Table 3: Reaction scale: a brief description (as with Table 1) should be included, and the reference used and adapted to create the scale (Christiansen et al.).
P5L157: (16) and Tidyverse. The line should start with Christiansen et al. and Tidyverse, instead of (16)….
Results:
P6L175: Figures 5 and 6 are mentioned, but not such figures are included in the document.
P6L170: Figure 3 in not mentioned in the results.
Discussion:
P7L185: safety of riders: there is an extra space between of and riders.
L7L193: Table 8 is mentioned, but there is no Table 8 in results. Should be Table 4?
P7L197: missing period between again and These
In my opinion, and knowing that there are only few available studies related to this topic, more discussion with previous studies should be included. The first two paragraphs are more like an analysis of the results, than a discussion. Some interesting papers like Hartmann et al. 2021, Marsboll and Christensen 2015, Christensen et al. 2021 could be included to discuss the present results.
Also, do the authors consider that there could be some breed effect in the difference between this study and Hanggi? Could be interesting to consider that factor, or age?
Author Response
Response to Reviewer 1
We have added additional references to our article per Reviewer 1's request and have removed repetitive sentences. The title has been changed to Evaluating the reaction to a complex rotated object in the American Quarter Horse (Equus caballus). Each of Reviewer 1's specific comments and line edits are listed below.
This is an interesting work that adds new information regarding reaction of horses, that will help a better and safe management.
Title: Equus should be in uppercase.
E in Equus changed to be capitalized.
Introduction: explains well the topic.
Thank you!
P2L55: in reference 4, the author´s last name should appear more than the reference number.
Introduction was rewritten.
P2L69: Additionally, A horse´s sudden flight reaction. A should be lowercase.
Introduction was rewritten.
P2L66-79: some aspects in this paragraph are mentioned several times. For example, the risk of accident with a spooky or scared horse is mentioned five times. In my opinion, that paragraph can be summarized in a central idea that is that reaction in horses or spooky animals can cause accidents and injuries for both riders/handlers and horses, and that those reactions are common in the equine industry.
Introduction was rewritten
In addition, in the previous paragraph (L61-63) it was already mentioned that horses led by a handler can change their behavior, similar to L76-77. That idea and study can be mentioned in the second paragraph.
Introduction was rewritten
Material and Methods:
Table 2: the title repeats Behavioral signs (P4L147)
Deleted the first “Behavioral Signs” in title
Table 3: Reaction scale: a brief description (as with Table 1) should be included, and the reference used and adapted to create the scale (Christiansen et al.).
Added description of table and appropriate reference
P5L157: (16) and Tidyverse. The line should start with Christiansen et al. and Tidyverse, instead of (16)….
Changed reference format to Christensen et al. and Tidyverse
Results:
P6L175: Figures 5 and 6 are mentioned, but not such figures are included in the document.
Changed to appropriate figures (Figures 2 and 3)
P6L170: Figure 3 in not mentioned in the results.
Figure 3 is now mentioned in the results.
Discussion:
P7L185: safety of riders: there is an extra space between of and riders.
Extra space deleted.
L7L193: Table 8 is mentioned, but there is no Table 8 in results. Should be Table 4?
Table 8 changed to Table 4.
P7L197: missing period between again and These
Period added.
In my opinion, and knowing that there are only few available studies related to this topic, more discussion with previous studies should be included. The first two paragraphs are more like an analysis of the results, than a discussion. Some interesting papers like Hartmann et al. 2021, Marsboll and Christensen 2015, Christensen et al. 2021 could be included to discuss the present results.
Thank you for the suggestion! These studies were very informative and fit well in the discussion to enhance it.
Also, do the authors consider that there could be some breed effect in the difference between this study and Hanggi? Could be interesting to consider that factor, or age?
Added following chunk to the discussion: “There are possible differences in personality and reactivity by breed (Lloyd et al 2008). The traits that showed the most variability between breeds were excitability and anxiousness (Lloyd et al 2008). This has been thought to be because of what each breed has been bred for over time. For example, Thoroughbreds tend to show a bigger flight response because they are bred to race and are expected to leave the gate quickly (McGreevy and Thomson 2006). Age has also been shown to have an effect on reactivity. In a study comparing horses ranging in age from two months to two years, younger horses were more reactive (Calviello et al 2016).”
Reviewer 2 Report
Dear Authors,
attached please find a pdf with comments.
Regards

Author Response
Response to Reviewer 2
Per Reviewer 2's request, the title of our article has been changed to "Evaluating the reaction to a complex rotated object in the American Quarter Horse (Equus caballus). Reviewer 2's specific comments and line edits are listed below.
Please include a specific horse breed in the title, the one you analyzed-American Quarter Horse.
suggested title: Evaluating the reaction to a complex rotated object in the American Quarter Horse.
Changed “domestic horse” to “ American Quarter Horse
Species names should begin with a capital letter and be written in italics.
New title: Evaluating the reaction to a complex rotated object in the American Quarter Horse (Equus caballus)
What does "a" mean?
Changed a to α
This list of references should be written with accordance to Animals Journal guidelines for authors.
Fixed references for Animals Journal guidelines
Two comment boxes did not have any text. I’m not sure if there were comments at Table 4 or Conclusion, but I am unfortunately unable to see them.
Reviewer 3 Report
Dear authors,
I have reviewed your manuscript. Please find my comments in the PDF file I send you attached.
I have divided my feedback into some general comments and then specific comments referring to single lines/contents of the paper.
I hope you find my comments helpful to revise your manuscript.
Best wishes,
your reviewer

Round 2
Reviewer 2 Report
The Authors have made the suggested improvements. I have no additional comments.
Regards
Author Response
The statement on ethics has been revised as well as other minor revisions.
Reviewer 3 Report
Dear authors,
Thanks for your work and addressing the comments I previously made!
Please find my comments on your second draft in the attached PDF.
Best wishes!

Author Response

(The authors gave the same response as above.)
